# Novel Techniques and Future Perspective for Investigating Critical-Size Bone Defects

**DOI:** 10.3390/bioengineering9040171

**Published:** 2022-04-11

**Authors:** Elijah Ejun Huang, Ning Zhang, Huaishuang Shen, Xueping Li, Masahiro Maruyama, Takeshi Utsunomiya, Qi Gao, Roberto A. Guzman, Stuart B. Goodman

**Affiliations:** 1Department of Orthopaedic Surgery, Stanford University, Stanford, CA 94304, USA; hhhhh@stanford.edu (E.E.H.); ningzzz@stanford.edu (N.Z.); hss2018@stanford.edu (H.S.); xuepli@stanford.edu (X.L.); masa.maruyama1460@gmail.com (M.M.); takeshiu0625@gmail.com (T.U.); qigao7@stanford.edu (Q.G.); guzmanr@stanford.edu (R.A.G.); 2Department of Bioengineering, Stanford University, Stanford, CA 94304, USA

**Keywords:** critical-size bone defect, mass cytometry, CyTOF, imaging mass cytometry (IMC), scRNA-seq, Luminex

## Abstract

A critical-size bone defect is a challenging clinical problem in which a gap between bone ends will not heal and will become a nonunion. The current treatment is to harvest and transplant an autologous bone graft to facilitate bone bridging. To develop less invasive but equally effective treatment options, one needs to first have a comprehensive understanding of the bone healing process. Therefore, it is imperative to leverage the most advanced technologies to elucidate the fundamental concepts of the bone healing process and develop innovative therapeutic strategies to bridge the nonunion gap. In this review, we first discuss the current animal models to study critical-size bone defects. Then, we focus on four novel analytic techniques and discuss their strengths and limitations. These four technologies are mass cytometry (CyTOF) for enhanced cellular analysis, imaging mass cytometry (IMC) for enhanced tissue special imaging, single-cell RNA sequencing (scRNA-seq) for detailed transcriptome analysis, and Luminex assays for comprehensive protein secretome analysis. With this new understanding of the healing of critical-size bone defects, novel methods of diagnosis and treatment will emerge.

## 1. Introduction

There are over 150 million new fractures globally every year [1]. When a substantial area of bone is lost or excised, and the gap is too large to heal spontaneously, the fracture does not unite and becomes a non-union; the gap in the bone is referred to as a “critical-size bone defect”. Approximately 100,000 cases of fractures in the USA result in nonunion annually [2,3,4] and the cost of medical care is estimated to be $2.5 billion [5]. There are multiple causes leading to nonunion, including trauma, infection, prior radiation, excision of a bone tumor, etc. The bone defect will not heal due to the size of the defect and disruption of the normal processes that facilitate osteogenesis and angiogenesis.

One of the current standards of care for treating a critical-size bone defect is mechanical stabilization to prevent motion at the defect site and to perform bone autogenous bone grafting [6]. However, this approach requires surgeries at several anatomic sites, and is associated with additional risks of infection and pain, etc. One alternative treatment includes bone transport surgery, which is extremely burdensome and painful for patients [7], and recovery from this surgery may take months to years. Biologic materials or strategies provide an alternative to the conventional options of treatments such as autogenous bone grafting. Novel biomaterial scaffolds, cellular therapies with stem cells, bone marrow aspirate, platelet-rich plasma (PRP), and BMP are currently available for use in the treatment of critical-size bone defects [8,9,10,11,12].

As the gap of the bone defect increases, the possibility of spontaneous bone healing decreases significantly, and is an unmet clinical need. Over the past decade, researchers have applied the principles of tissue engineering, regenerative medicine, and cell therapy for healing bone defects. In this regard, mesenchymal stem cells (MSCs) have been demonstrated to play a pivotal role in bone healing [13,14]. However, crosstalk among many different types of cells collectively, including cells of the mesenchymal stem cell-osteoblast lineage, the monocyte-macrophage-osteoclast lineage, and the endothelial cell lineage, contribute to the bone healing process [15]. At present, little is known about the exact composition and the detailed crosstalk of the cell populations that direct the bone healing activities. 

Traditional methods of analysis such as micro-computational tomography (µCT), biomechanical testing, histomorphometry, histochemical and immunohistochemical staining, enzyme-linked immunosorbent assay (ELISA), mRNA analysis using polymerase chain reaction (PCR), and others may be insufficient in answering fundamental questions concerning the regeneration of lost bone. Recently, more sophisticated technologies and methods of analysis have emerged to better understand the biologic principles and mechanisms of bone healing. In this review, we first compare several reliable and validated models for research in critical-size bone defects; then we discuss innovative state-of-art technologies that may facilitate and advance the current research. Finally, we will elucidate future directions of potential therapeutic interventions to heal critical-size bone defects.

## 2. What Are the Current Models to Study Critical-Size Bone Defects?

Numerous models of critical-size bone defects have been used to evaluate the efficacy of bone graft substitutes. These models can be classified into four major types, including segmental long bone defects, calvarial defects, partial cortical defects, and cancellous bone defects (drill holes) [16]. Of these, segmental long bone and calvarial defect models have been widely used for basic research on bone healing of critical-size defects [16] (Figure 1, middle). Below, we summarize the most commonly used animal models of critical-size long bone and calvarial defects (Table 1 and Table 2).

Small animal models (mice, rats, and rabbits) are commonly used for fundamental studies or screening experiments because of the lower cost and greater ethical acceptance. In addition, immunodeficient mice and rats have been used when researchers intended to apply heterogeneous resources (human mesenchymal stem cells, human platelet-rich plasma, human bone marrow concentrate, etc.) into the critical-sized bone defect in animals [17,18,19,25,26,27]. Immune competent animals can also be applied when the therapeutic intervention is from homologous species (autograft or syngeneic graft). Compared to small animals, large animals (dogs, pigs, goats, and sheep) have advantages in their similarity to humans including anatomical, physiological, and biomechanical properties, as well as the surgical procedures to create defects; large animals are suitable for more advanced preclinical experiments [50], however, they are expensive and require more time and effort for their overall care.

Mice and rats are commonly used as models for a femoral or calvarial critical-size bone defects. Rabbits have been used in a model of radial or ulnar bone deficiency [58]. In large animals, sheep are the often used large animal species for a model of the critical-size long bone defect, most often in the tibia [49,58]. Dogs, goats, and pigs are also used, but their long bones are relatively shorter compared to human long bones [50,58].

Segmental critical-sized femoral and tibial bone defects require stabilization using a fixation technique e.g., an external fixator, a plate and screws, or an intramedullary nail. Interestingly, in a model of segmental critical-sized radial and ulnar bone defects, bone defects can be stabilized without any fixation device because the radius and ulna are firmly linked together by the interosseous membrane [16,59]. In addition, a model of calvarial bone defects does not require any method of stabilization.

Long bones and calvarial bones are very different from each other [56]. A long bone is derived from the lateral plate mesoderm and is primarily formed via endochondral ossification, whereas the calvarium is a flat bone, which is derived from cephalic mesoderm and neural crest and is mainly formed via membranous ossification. The vascular supply to these bones is distinctive as well. Long bones have three sources of blood supply: nutrient vessels, metaphyseal and epiphyseal vessels, and periosteal vessels [60]. The nutrient vessels enter the bone through their respective foramina and reach the medullary cavity, and they distribute branches and supply blood to the medullary cavity and cortical bone; the periosteal vessels supply the outer cortical bone [34,37]. On the other hand, the blood supply to calvaria is delivered by periosteal vessels. At the microvasculature of flat bones with less than 0.4 mm thickness in the rat (e.g., thinnest parts of scapula), the bone tissue is supplied by only the periosteal and dual network [36].

Thus, in the choice of an animal model of a critical-size bone defect, the type of bone defect suitable for the research question to be explored should be considered. In addition, the animal species, anatomical site, defect size, and fixation methods as well as age and sex should also be considered based on the clinical relevance of each research. For example, older animals can shed light into the pathophysiology of elderly patients with osteoporosis. Moreover, observation periods should be adapted to the type of bone defect and/or animal model: smaller defects and defects in young or smaller animals repair faster than larger defects, especially those in older or larger animals [35].

## 3. How Do Novel Techniques Facilitate Current Research?

To study the critical-size bone defect in the aforementioned animal models, multiple traditional methods have been used by many research groups to reveal important cellular, spatial, transcriptional, and translational information. For example, Marietta et al. used flow cytometry to study cell mobilization in the rat bone defect model [61]; Ueno et al. applied immunohistochemistry (IHC) for the macrophage and MSCs phenotype analysis in a murine long bone critical-size defect model [19]; Shi-Cong et al. explored possible molecular mechanisms of the therapeutic effects of bone regeneration on critical-size bone defects [62]; Huanxin et al. employed ELISA to verify the concentration of growth factors that facilitate bone regeneration in a rabbit bone defect model [63]. Given the above scientific advances, the traditional methods used in these studies can only observe a limited number of biomarkers or parameters and constrain the development of more in-depth insights and more comprehensive views towards their study objects. Therefore, we are recommending the following techniques to further our understanding of the biology of critical-size bone defects.

### 3.1. Mass Cytometry (CyTOF) for Enhanced Cellular Analysis (vs. Flow Cytometry)

For almost half a century, classical flow cytometry has been widely used for phenotyping cell populations and investigating signaling pathways and cytokine secretion [64]. However, the excitation and emission spectra of the fluorophores often overlap with one another. To achieve the optimal performance of flow cytometry, complicated compensation processes need to be added to minimize the spillover effect between different fluorophores. Thus, the maximum number of fluorophores being used simultaneously must be limited. The conventional flow cytometer usually achieves its optimal performance with less than 10 fluorophores, i.e., quantification of 10 different biomarkers simultaneously. Recently more advanced flow cytometers with higher resolution (such as BD FACSymphony, Cytek Aurora) have been developed to enable the simultaneous application of over 20 fluorophores. However, designing a large panel of available antibodies with compatible fluorophores still remains challenging. In addition, isotype controls and Fluorescence Minus One (FMO) controls are always needed to be included in the experiments [65], which further complicates the technique, especially when designing a large panel of antibodies.

Mass cytometry is a technology that combines mass spectrometry and flow cytometry. Following the basic principle of flow cytometry, mass cytometry incorporates the precision capability of mass spectrometry to quantify over 40 cellular parameters simultaneously [66]. The current instrument to perform mass cytometry was developed in 2009 by researchers at the University of Toronto and is called “Cytometry by Time-Of-Flight”, or “CyTOF” [67]. Instead of using fluorophores, mass cytometry uses heavy stable metals (primarily the lanthanide metal family) as distinct labels for multiparametric single-cell analysis (Figure 1, up left). Since there are more than 40 different metals and isotopes, and each with distinct atomic weights that can be distinguished by mass spectrometry, CyTOF enables mass cytometry to simultaneously measure over 40 parameters in a single experiment [68]. More importantly, unlike the wide spectra of the fluorophores, the signals generated from mass spectrometry are unique, with little overlap between different metals, which significantly improves the reliability and accuracy of the data.

During the healing of bone defects, mesenchymal stem cells (MSCs) are known to be the precursors of osteoblastic lineage cells [69]. However, MSC lineage cells are not the only cells involved in this activity; crosstalk among many different types of cells collectively contributes to the bone healing process [15]. In fact, there is limited information concerning the exact cellular constituents and complex crosstalk among different cell populations that direct the process of bone healing. To address this critical question, mass cytometry represents a powerful tool to demonstrate the crosstalk among multiple cell types as well as elucidate the activation of various signaling pathways and cytokine secretion at the single-cell level.

Here we use an example to demonstrate how mass cytometry can reveal information on macrophage heterogeneity that normally we would not be able to visualize using the classic method of flow cytometry (Figure 2).

We collected samples from femoral bone defects that have been transplanted with bone graft for 7 days to analyze the bone healing process. The samples were analyzed using mass cytometry with a 40-antibody panel and the data is visualized with SPADE (Spanning-tree Progression Analysis for Density-normalized Events) to explore the cellular heterogeneity within each cell population (Figure 2). We annotated the major cell populations on the SPADE tree and verified their credibility on a viSNE plot (see the surrounding contour viSNE plots). In SPADE, cells with similar attributes will cluster into nodes and based on their similarities, different nodes will form a minimum-spanning tree. Although macrophages consist of less than 10% of the total cell population in the defect at 1 week, they have significantly more nodes (cell clusters) on this SPADE tree than any other cell type. Macrophages are reported to have a major immunomodulatory effect during bone healing [15], and they are normally categorized into M0 undifferentiated, M1 pro-inflammatory, and M2 anti-inflammatory pro-reconstructive macrophages using flow cytometry with limited number of phenotypical markers. The mass cytometry data in Figure 2 demonstrates that macrophages at the bone defect sites, are an extremely diverse cell population with many sub-populations that function collectively during the bone healing process. 

As an emerging technology, mass cytometry has been applied primarily in human studies, focusing on complicated cellular composition of primary human samples, such as whole blood, bone marrow, dissociated tissues, etc. [66,70,71,72,73]. Among the few studies on animal models other than humans, the majority of studies are focused on peripheral blood or bone marrow [74,75], probably due to the availability and accessibility of the cells. Clearly, it would be challenging to harvest and isolate cells from critical-size bone defect models in order to elucidate the cellular crosstalk during bone healing.

Given the large multiparametric power that mass cytometry possesses, this technology also has some limitations.

First, the setup of mass cytometry requires a completely different set of instruments compared with flow cytometry. This results in high upfront costs and hinders the wide application of this technology.Second, to have this multiparametric power under control, mass cytometry also requires personnel who are leading the project to be well-trained and experienced in this field in order to achieve reliable and consistent outcomes.Finally, as the cells are vaporized and atomized during the process, no living cells will be recovered from mass cytometry for any other downstream analysis.

### 3.2. Imaging Mass Cytometry (IMC) for Enhanced Tissue Imaging (vs. IHC)

While cytometry (both flow cytometry and mass cytometry) is a useful technique for defining the properties of individual cells present in the tissues in bone defects or in circulating cells in the blood, the cells in the harvested tissue must be dissociated. In doing so, the spatial relationships of the cells within the tissues are lost. Thus, the contextual relationships and interactions among different cell types is difficult to determine without additional histological or other techniques.

Imaging Mass Cytometry (IMC) is the visualization of the results of mass cytometry [76,77]. Instead of measuring the metal-labeled parameters in cell suspension, IMC uses a pulsed laser to ablate metal-labeled cells on a slide and then a stream of inert gas carries the vaporized cells in sequence into the mass cytometer for analysis. Each ablation spot represents one pixel on an image, which shows the information of one piece of the cell. Once the information on every ablation spot has been collected, the reconstructed images are generated to reflect the distribution of the cells in situ (Figure 1, up right). 

If mass cytometry is regarded as an upgraded version of flow cytometry, then IMC is an upgraded version of IHC (immunohistochemical staining). Similar to mass cytometry compared with flow cytometry, the strength of IMC over IHC is the capability of visualizing over 40 parameters simultaneously, with minimum signal contamination. Moreover, compared to mass cytometry, IMC gives the location of cells in addition to the multiplexed mass cytometry data, which underlines the importance of cell-cell interactions in situ.

IMC is a very recent technology developed in 2014 [77] This technology has been utilized in only about 60 publications and the vast majority of these are human studies [78,79,80,81,82]. For human tumor samples where the cell-cell crosstalk and immune microenvironment is extremely complex, the potential of this powerful imaging technology can be fully exploited. Very few studies have leveraged the power of this IMC technology for bone research [83], especially in mice. IMC represents an enhanced alternative to IHC and can detect up to 40 different parameters on the very same image, making it possible to decipher the interactions among multiple cell types.

Here, we show an example of using IMC imaging to reveal cell–cell spatial infor-mation in the human synovial membrane (Figure 3). We prepared the IMC slide similar to that for IHC. Upon staining, traditional IHC may only use a combination of 3–4 an-tibodies to show limited cell types and thus, the resultant information is limited. Here, we stained the slide with a 30-antibody panel for IMC imaging, and seven antibodies were chosen to visualize the major cell types, which include fibroblasts, epithelial cells, M1 and M2 macrophages, B cells, and T cells simultaneously on the same slide (Figure 3). Together with other cell subtypes or functional markers, this IMC image can allow us to conduct highly multiparametric analysis to reveal a large amount of information on the human synovial membrane.

While IMC is such a powerful tool to dissect the in situ detailed cell–cell crosstalk at the single-cell level, it has limitations:First, as an extension of mass cytometry, CyTOF is required to perform IMC. Furthermore, it also requires an additional machine called “Hyperion”, where cells go through laser ablation to generate metal particles streams and send the information to CyTOF for analysis. Thus, the upfront investment for instruments is even steeper.Second, despite similar basic principles, the antibodies used for mass cytometry and for IMC are not interchangeable. This means that to perform IMC, a completely new panel of 30+ antibodies must be designed and developed, which leads to additional cost and time.Third, similar to mass cytometry, IMC requires well-trained personnel who not only need to understand IMC, but also know the details of mass cytometry, in order to achieve the best performance of this technology.

Although these drawbacks hinder the wider application of IMC, the future of IMC is promising. As this powerful technology continues to evolve, more researchers and institutes utilize this technology to solve challenging scientific questions. More applications of this technology will also drive down the overall cost, further making it accessible to greater numbers of the scientific community.

### 3.3. Single-Cell RNA Sequencing for Detailed Gene Expression Analysis

To further explore the biological mechanisms of the healing process on critical-size bone defect, it is essential to understand the transcriptome (the mRNA in the cell) to deconvolute or simplify the function of the genome and thereby, understand the molecular constitution of the cell. Hybridization-based microarrays were first applied to quantify high-throughput gene expression with fluorescently labelled complementary DNA (cDNA) techniques, however these methods demonstrated high background levels and difficulties in cross-comparison [84]. In contrast, RNA sequencing (RNA-seq) offers a more precise measurement with more transcript complexity [85].

Since the advent of the next-generation sequencing technology in the mid-2000s, RNA-seq has become a powerful tool to reveal the whole genome expression level using diverse study models. For studies in bone, RNA-seq has been utilized to evaluate the skeletal distribution and gene expression of various skeletal cells [86]. 

Nevertheless, the cell populations within the bone are extremely diverse. A thorough understanding of the mechanisms of bone formation demands the clarification of the specific role of each skeletal cell, including osteoblasts, osteoclasts, osteocytes, in addition to their crosstalk with immune cells [15]. In addition, the high degree of plasticity and phenotypic instability of MSCs, which are indispensable in generating and repairing skeletal tissues, can cause in vitro studies to be less reproducible among different laboratories [87]. Also, isolating defined cells from multiple subpopulations using one single marker is a formidable challenge [88,89]. As the single-cell technologies become more powerful and accessible, single-cell RNA sequencing (scRNA-seq) is regarded as a pivotal complementary method to reveal mixed cells and identify discrete subpopulations contributing to bone physiology, as well as differentiation hierarchies for bone marrow stromal cells and critical transcription factors (Figure 1, bottom left).

Bolander et al explored the mechanism of serum-free preconditioning human periosteum-derived stem cells (hPDSC) on critical size tibial defects in mice and demonstrated a phenotypic shift to a more improved in vivo bone regeneration at the single cell level. Besides strong evidence of ectopic and orthotopic bone forming capacity, scRNA-seq also elucidated activation of pathways and transcriptional regulators involved in bone defect healing [90].

There are mainly two approaches to capture single cells: one is based on methods that rely on index sorting by FACS; the other is the technique utilizing a microfluidics-based capture of cells into droplets [91]. Recently, several groups have developed index-based techniques centered on the initially emerging method known as SMART-seq, which relies on sorting single cells into individual wells of 96 or 384-well plates [92]. SMART-seq2 is the improved version of SMART-seq, featuring the generation of full-length cDNAs and thereby allowing the detection of gene isoforms. This method is suitable for experiments that deal with rare cell populations and allows a larger number of genes to be detected in each individual cell; however, this method may not be the most efficient way for experiments that require thousands of individual cells. Conversely, recently developed droplet/microfluidics-based methods, including Drop-seq, inDrop, and 10× Genomics platform, allow high throughput, 10,000 to 100,000 cells to be captured and processed in a rapid fashion [93]. While 10× Genomics Chromium system is the most cost- and time-effective method, this platform is currently capable of detecting 500–1500 genes per primary cell. With the improvements of these methods, it is critical for the musculoskeletal scientists to select an appropriate platform, and this largely depends on the biological question to be addressed. In order to obtain meaningful results, a reasonable estimate of the level of cellular heterogeneity should be calculated prior to conducting the experiment, which can be challenging.

However, there are some limitations with this advanced method. First, the cell isolation step might be challenging. Large cells, such as osteocytes and osteoclasts, can be physically disrupted or lost, thus leading to biased results in determining cell frequencies. Instead, sequencing of single nuclei represents an alternative solution [94]. Endocortical cells were first discovered by single nuclei sequencing, which revealed the environmental and pharmacologic perturbations of osteoblasts.

In addition, enzymatic digestion needs to be optimized to ensure cell viability. This isolation-associated procedure can induce extra stress responses in cells and cause transcriptional changes, which may confound the subsequent analysis.

The cost of conducting scRNA-seq is dependent on the library preparation protocols and targeted sequencing depth. Regarding the library preparation, by using a droplet-based library construction protocol, the analysis would cost around $0.1 per cell, while full-length protocols could cost around $25–30 per cell [95]. In general, more sensitive protocols (e.g., SMART-seq2) are required with greater sequencing depth to obtain whole-transcript coverage, which is more expensive than droplet-based protocols (e.g., Drop-seq) with lower sequencing depth. Applying full-length library preparations and high sequencing depth with a large number of cells could significantly increase the cost of the experiment.

Furthermore, while multiple data analysis tools are available, the analysis pipelines are yet to be standardized, especially for identifying novel cell populations and changes in cell type-specific gene expression. A thorough understanding on the limitation of scRNA-seq can help us make an informed decision and facilitate the exploring process of the complexity of musculoskeletal disease.

### 3.4. Multiplex Assays for Comprehensive Protein Panel Analysis 

To fully understand cell and tissue function in a mechanistic way, one must determine what proteins, peptides, glycoproteins and other such substances are being produced by biological samples. Enzyme Linked Immunosorbent Assay (ELISA) is one such method, first described in 1971 by Engvail and Perlmann [96]. ELISA is an immunoassay technique in which antibodies attach to a targeted molecule; the complex is then detected and quantitated by one of several different methods. ELISA was the mainstay of analysis for many years, but proved to be labor intensive, costly, and originally restricted to identifying a single target molecule. Techniques were needed to perform multiple assays for peptide-based substances in a faster, more efficacious, reproducible, and cost-effective way. 

High-throughput assay technologies enable scientists to conduct multiple measures in a single run. Luminex is one of these high-throughput assays that measures multiple analytes for a comprehensive protein panel in one sample (Figure 1, bottom right). The Luminex xMAP® (x- Multi-Analyte Profiling) technology uses labeled microspheres or beads. Because of the small size and low density of the microspheres/beads with multiple formats, including magnetic and non-magnetic, Luminex xMAP® can simultaneously capture multiple nucleic acid- and protein-based analytes (3–500 targets) from a single run, by reading the microspheres/beads individually using an Luminex xMAP® instrument. First reported in 1997 [97,98], Luminex xMPA® technology has become a cutting edge technique for clinical diagnosis and basic research [99]. The first application of Luminex was approved by the FDA in 2001, making it the first FDA-approved technology of multi-factor analysis for clinical diagnosis.

Luminex resolved the limited surface and limited throughputs of ELISA in the plate-based 2D system. As mentioned previously, with the help of the microspheres or beads, the surface used for detection was enlarged from a 2D to a 3D system resulting in higher throughputs. Luminex also can provide high sensitivity detection with low volumes of samples. Compared with ELISA, Luminex is time- and labor-saving and low-cost.

Luminex is also popular in both pre-clinical and clinical research. In orthopaedic research, Luminex is a novel and powerful tool for profiling the biomarkers or cytokine/chemokine levels associated with orthopaedic diseases, injuries, and treatments such as MSC-based therapies [100,101,102]. Luminex assay is also used in the research field of critical size bone defects. Johnson et al. [103] employed Luminex assay to detect the in vivo cytokines expression of a critical-size segmental bone defect model. The serum protein level was quantified by Luminex assay in a murine critical size bone defect model [104]. In studies of critical-size bone defects, Luminex assays could potentially identify the presence and production and overall contribution of specific proteins and related molecules at the bone defect sites. Luminex assay is also useful for pathogen detection, making it a powerful tool, for example, to detect and analyze the presence of infection after prosthesis implantation in arthroplasty or other orthopaedic surgical procedures [105]. Luminex assays can facilitate the diagnosis, prevention, and treatment of early pathological processes, and can be used for drug discovery focused on developing specific therapeutic targets and therapeutic strategies.

## 4. Future Direction and Conclusions: Translation of Findings from Bench to Bedside 

This review summarized currently used animal models for examining critical-size bone defects and highlighted emerging techniques to investigate the biological mechanisms underlying these defects. The choice of appropriate animal model is based on the question being asked, in light of the potential for the new strategy, technique or therapeutic to be translated from bench to bedside [106]. Traditional techniques such as flow cytometry, IHC, ELISA and bulk RNA-seq have been widely used to explore the progression of healing of bone defects for quite some time; with the novel techniques introduced in this review, including CyTOF, IMC, scRNA-seq and Luminex assays, a more detailed and comprehensive assessment of healing of critical-size bone defects over time can be effectively elucidated at the cell and tissue, protein, and molecular levels.

Bone formation and bridging is the obvious endpoint for evaluating the utility and efficacy of treatments for critical size bone defects. The processes by which cells perform their biological tasks in accomplishing this goal will undoubtedly guide the development of new therapies or novel biomaterials for bone defect repair.

After harvesting and processing the samples at the bone defect site, the novel techniques can be used for the analysis of interest. Using CyTOF, the presence and changes of the different cell populations of interest can be identified by the established panel of biomarkers. Using IMC, the cell localization and cell-cell interaction information can be demonstrated. The details at the cell and tissue level can be revealed using a combination of CyTOF and IMC. Besides checking the cell population changes and the potential signaling pathways involved in the process, the relevant protein changes such as the secretom profiles can also be analyzed by using Luminex. In addition, scRNA-seq could be done to show the gene expression profile of the cell populations of interest. 

There are still some limitations to each of these new techniques. Generally, the high upfront cost and the requirement of professional personnel for data analysis slows down their wider applications. Nevertheless, the potential benefits still overweigh the obstacles. The integration of these multiple techniques will bring novel and in-depth insights into this research field for understanding and treating critical-size bone defects. With the help of these powerful tools, we can not only understand the biological processes in more detail, but also develop novel strategies including biomaterial-, cell-, cytokine-, chemokine- and gene- based therapeutical methods. That approach will facilitate the translation of these preclinical technologies into clinical practice.

## Figures and Tables

**Figure 1 bioengineering-09-00171-f001:**
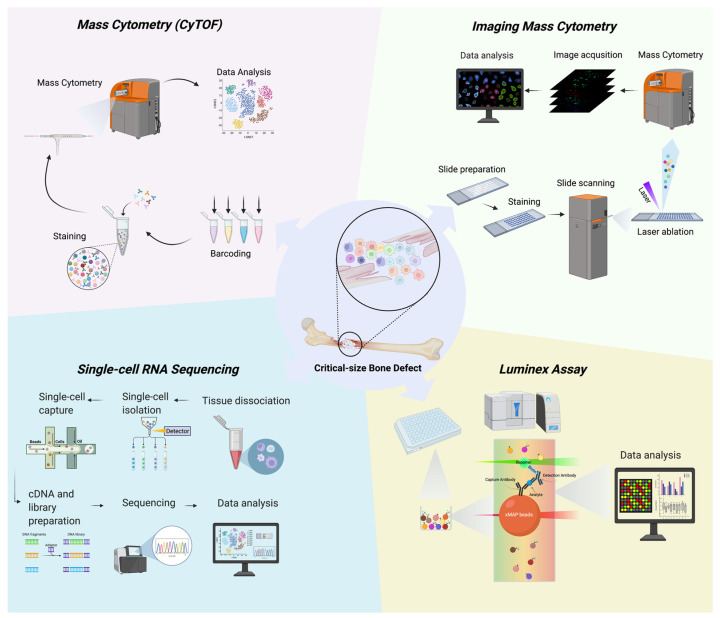
Novel techniques for investigating critical-size bone defects.

**Figure 2 bioengineering-09-00171-f002:**
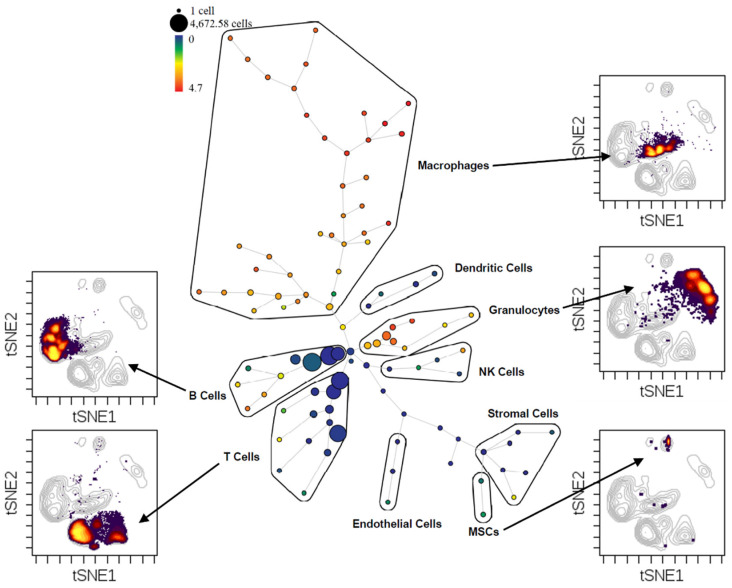
SPADE tree with contour viSNE plots to show heterogeneity of different cell populations within the critical-size bone defect site.

**Figure 3 bioengineering-09-00171-f003:**
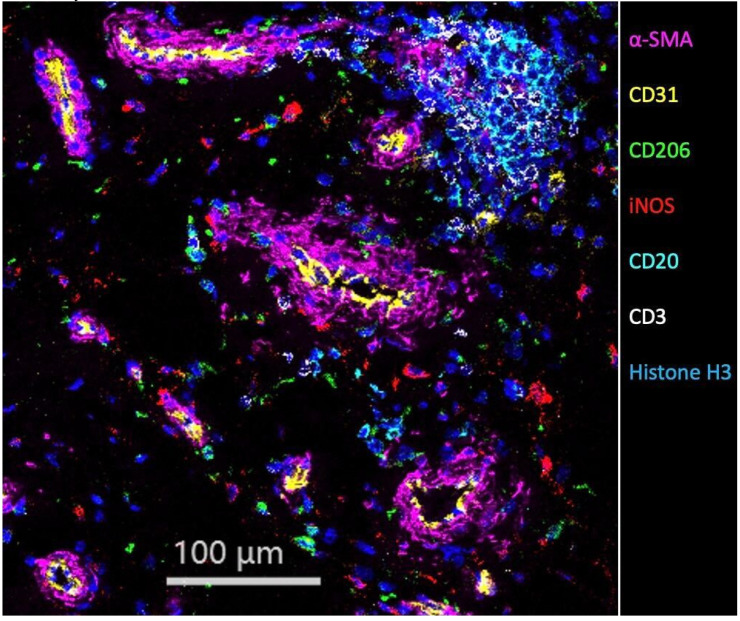
IMC imaging on human synovial membrane sample. Seven markers from a 30-marker panel were shown in the image. Magenta: α-Smooth muscle actin (α-SMA); Yellow: CD31; Lime: CD206; Red: iNOS; Cyan: CD20; White: CD3; Blue: Histone H3.

**Table 1 bioengineering-09-00171-t001:** Animal models of the segmental critical-size bone defect.

Species	Defect Site	Defect Size	Fixation Techniques	References
Mouse	Femur	2–5 mm	External fixator Plate and screw Intramedullary pinning	[17,18,19,20,21,22]
	Tibia	3 mm	External fixator Intramedullary pinning	[23,24]
Rat	Radius	5 mm	Without stabilization	[25,26]
	Femur	4–20 mm	External fixator Plate and screw Intramedullary pinning	[27,28,29,30]
	Tibia	3–8 mm	Intramedullary pinning	[31,32,33]
Rabbit	Radius	>14 mm	Without stabilization	[34]
	Ulna	15–20 mm	Without stabilization	[35,36,37]
	Femur	10–15 mm	Plate and screw Intramedullary nail	[38,39,40]
	Tibia	10–15 mm	Plate and screw Intramedullary nail	[41,42]
Dog	Radius	20 mm	Without stabilization	[43]
	Ulna	20 mm	Without stabilization	[44]
	Femur	20 mm	Plate and screw	[45]
	Tibia	6 mm	External fixator	[46]
Pig	Femur	30 mm	Plate and screw	[47]
	Tibia	30 mm	Plate and screw	[48]
Sheep	Tibia	30–40 mm	External fixator Plate and screw Intramedullary nail	[49,50,51,52,53]
Goat	Tibia	30 mm	Plate and screw	[54]

**Table 2 bioengineering-09-00171-t002:** Animal models of the calvarial critical-size bone defects.

Species	Defect Size	References
Mouse	3–8 mm	[55,56]
Rat	5–8 mm	[55,56,57]
Rabbit	8–15 mm	[56]
Dog	20 mm	[55,56]
Pig	10 mm	[55]
Sheep	10–30 mm	[55,56]

## Data Availability

The data presented in this study are available on request from the corresponding author. The data are not publicly available due to confidentiality reason of related research project.

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
