# Peer review of "Novel Techniques and Future Perspective for Investigating Critical-Size Bone Defects"

_bioengineering, 2022, doi:10.3390/bioengineering9040171_

Round 1
Reviewer 1 Report
I read carefully the manuscript entitled “ Novel Techniques And Future Perspectives For Investigating Critical-Size Bone Defects”. In this work, the authors made a review of cutting edges method that may be useful in the research domain of bone regenerative medicine especially in the context of critical-size bone defects.
The article is well written and the point is quite clear. However, I have some questions and remarks to improve this work for publication.
Questions:
- General consideration: the new technologies described in this paper are not yet widely used in the field of bone regenerative medicine. As a proof, very few paper using these technologies are cited, most of them are mostly from cancer, immunology research. The author should insist on this point in the introduction and explain that they are developing cutting-edge technologies that would be of high interest for the domain (as it is clearly explained in the title).
- 1: Introduction
- The cost of medical care is estimated to be $1.2 billion annually in the USA, the figures are from 2007, does the cost per patient is the same nowadays ?
- The cited reference in line with the gold standard treatment of critical-size bone defects was published in 2004. Has this treatment evolved since then? Can the authors cite alternative solutions currently used such as biomaterials, even if they are not as effective as gold standard treatment ?
- Mesenchymal stromal cells, mentioned in l44, are referred to as mesenchymal stem cell-osteoblast line in l46, thank you to the authors to homogenize the terms or to explain the choice of different terminologies for these cells
- 2: What are the current models to study critical-size bone defects ?
- All the §3 is devoted to new technologies that can facilitate/enhance the possibilities to analyze biological/physiological phenomena in the context of critical-size bone defects.
Consequently, I think that the authors should divide the §2 in two subparts: in the first part, will be conserved the current description of existing animal models for critical-size bone defect. In a second subpart would be described the current methods in use to study the healing of this critical-size bone defects according to the research strategy followed. This would make the §3 more in line with the §2. This would also resolve the first remark I made.
- 3: How do novel techniques facilitate current research ?
-sub-part 3.1: The authors should redefine the scope of flow cytometry, which requires isolated cells and therefore applies to dissociated tissue or circulating (blood) cells that is a very specific handling of tissues derived from volution of the bone defects created in animal models.
- sub-part 3.2: It would be very informative to add a figure presenting an example image resulting from Imaging Mass Cytometry so that readers unfamiliar with the technology can visualize the type of results that can be obtained..
- sub-part 3.2: the citing papers in relation with Imaging Mass Cytometry are in relation with tumor biology or immunology. The authors wrote in l212 that “very few studies have leveraged the power of this IMC technology for bone research, especially in mice(…)”. The author should cite these works as references, it would be for high interest for readers.
- sub-part 3.4: To my knowledge, Luminex is a trademark (referred as Flow metrics system in the cited literature). There is exist any generic denomination or any concurrence to this system ? Can the authors precise this point ?
- Is there any work using the Luminex technology in the field of critical-size bone defects apart from the cited example of detection of infection ?
Author Response
Please see the attachment for the revised manuscript. Thank you!
Questions:
- General consideration: the new technologies described in this paper are not yet widely used in the field of bone regenerative medicine. As a proof, very few paper using these technologies are cited, most of them are mostly from cancer, immunology research. The author should insist on this point in the introduction and explain that they are developing cutting-edge technologies that would be of high interest for the domain (as it is clearly explained in the title).
Response: Please see the response we’ve written in #2 below, which, as the reviewer pointed out, will also address this question altogether.
- 1: Introduction
- The cost of medical care is estimated to be $1.2 billion annually in the USA, the figures are from 2007, does the cost per patient is the same nowadays?
Response: We updated this part based on recent research data.
- The cited reference in line with the gold standard treatment of critical-size bone defects was published in 2004. Has this treatment evolved since then? Can the authors cite alternative solutions currently used such as biomaterials, even if they are not as effective as gold standard treatment?
Response: Autogenous bone grafting is still the most widely used “gold standard” treatment option used for critical size bone defects. We have updated this part of the manuscript, adding some alternatives to this conventional option such as the use of biomaterials and cellular therapies.
- Mesenchymal stromal cells, mentioned in l44, are referred to as mesenchymal stem cell-osteoblast line in l46, thank you to the authors to homogenize the terms or to explain the choice of different terminologies for these cells
Response: We have homogenized the terms in this manuscript and have used mesenchymal stem cell as the formal name for MSC.
- 2: What are the current models to study critical-size bone defects?
- All the §3 is devoted to new technologies that can facilitate/enhance the possibilities to analyze biological/physiological phenomena in the context of critical-size bone defects.
Consequently, I think that the authors should divide the §2 in two subparts: in the first part, will be conserved the current description of existing animal models for critical-size bone defect. In a second subpart would be described the current methods in use to study the healing of this critical-size bone defects according to the research strategy followed. This would make the §3 more in line with the §2. This would also resolve the first remark I made.
Response: This is a valid point! Since §2 discusses different models to study critical-size bone defects and §3 discusses different techniques, instead of dividing §2 into two parts, we have added a paragraph in the very beginning of §3 to discuss the current methods in use. This paragraph will lead to the discussion of these advanced techniques in §3 compared to the current methods in use.
- 3: How do novel techniques facilitate current research?
-sub-part 3.1: The authors should redefine the scope of flow cytometry, which requires isolated cells and therefore applies to dissociated tissue or circulating (blood) cells that is a very specific handling of tissues derived from volution of the bone defects created in animal models.
Response: You raised an important point. We believe this limitation of a lack of spatial relationships among cells exists in both flow cytometry and mass cytometry. Therefore, we added a paragraph to the beginning of the IMC section (sub-part 3.2) to discuss this limitation of cytometry and leads to the discussion of IMC.
- sub-part 3.2: It would be very informative to add a figure presenting an example image resulting from Imaging Mass Cytometry so that readers unfamiliar with the technology can visualize the type of results that can be obtained.
Response: We have added a figure presenting an example of an IMC image to address the reviewer’s comment.
- sub-part 3.2: the citing papers in relation with Imaging Mass Cytometry are in relation with tumor biology or immunology. The authors wrote in l212 that “very few studies have leveraged the power of this IMC technology for bone research, especially in mice (…)”. The author should cite these works as references, it would be for high interest for readers.
Response: We have added a reference which uses IMC technology for bone research that has published in 2020 to address this concern.
- sub-part 3.4: To my knowledge, Luminex is a trademark (referred as Flow metrics system in the cited literature). There is exist any generic denomination or any concurrence to this system? Can the authors precise this point?
Response: You are correct; “Luminex” is a trademark that is now owned by Diasorin Company (https://www.luminexcorp.com/). But the “Luminex multiplex assay” which is driven by xMAP technology has been licensed to several companies. So Luminex now is not only a trademark but also a technique based on the Luminex system. Luminex is now one of the most popular and widely used multiplex assays. We pointed out that “Luminex is one of these high-throughput assays that measures multiple analytes for a comprehensive protein panel in one sample”, and we discussed the application of Luminex as one of the most popular multiplex assays in the research of Critical size bone defect. We updated the subtitle of this part to “Multiplex assays for comprehensive protein panel analysis”.
- Is there any work using the Luminex technology in the field of critical-size bone defects apart from the cited example of detection of infection?
Response: We have added some reports using Luminex in the field of critical size bone defects in this part.

Reviewer 2 Report
The review manuscript hast two major focuses: animal critical size bone defect models and modern technologies, which could be applied to investigate the bone healing in these models.
The review is good structured and well written. Some minor improvement could be applied:
Introduction line 29 – reference is missing
Introduction lines 51-60 –microCT and biomechanical testing, which are commonly used to evaluate bone healing, should be mentioned as well.
Future direction & conclusion: At this part of the text, I believe it is important to mention the high costs of reviewed modern techniques, which limit its application until now. In addition, the importance of multi-disciplinary research team and professionally performed bioinformatics should be pointed out.
Author Response
Please see the attachment for the revised manuscript. Thank you!
The review manuscript hast two major focuses: animal critical size bone defect models and modern technologies, which could be applied to investigate the bone healing in these models.
The review is good structured and well written. Some minor improvement could be applied:
Introduction line 29 – reference is missing
Introduction lines 51-60 –microCT and biomechanical testing, which are commonly used to evaluate bone healing, should be mentioned as well.
Response: We have added these details and updated this part of the manuscript.
Future direction & conclusion: At this part of the text, I believe it is important to mention the high costs of reviewed modern techniques, which limit its application until now. In addition, the importance of multi-disciplinary research team and professionally performed bioinformatics should be pointed out.
Response: We have expanded the conclusion part to include some discussion about the limitations of these modern techniques, including the high costs as well as the requirement of professional personnel to handle these techniques.

Round 2
Reviewer 1 Report
All my comments and questions have been answered. I recommend accepting this manuscript for publication.
Reviewer 2 Report
All my comments to the manuscript are answered, I suggest the acceptance of manuscript for publication.